# MULTI-MODAL GRAPH NEURAL NETWORKS FOR LOCALIZED OFF-GRID WEATHER FORECASTING

## ABSTRACT

Urgent applications like wildfire management and renewable energy generation require precise, localized weather forecasts near the Earth's surface. However, weather forecast products from machine learning or numerical weather models are currently generated on a global regular grid, on which a naive interpolation cannot accurately reflect fine-grained weather patterns close to the ground. In this work, we train a heterogeneous graph neural network (GNN) end-to-end to downscale gridded forecasts to off-grid locations of interest. This multi-modal GNN takes advantage of local historical weather observations (e.g., wind, temperature) to correct the gridded weather forecast at different lead times towards locally accurate forecasts. Each data modality is modeled as a different type of node in the graph. Using message passing, the node at the prediction location aggregates information from its heterogeneous neighbor nodes. Experiments using weather stations across the Northeastern United States show that our model outperforms a range of data-driven and non-data-driven off-grid forecasting methods. Our approach demonstrates how the gap between global large-scale weather models and locally accurate predictions can be bridged to inform localized decision-making.

## 1 INTRODUCTION

In recent years, machine learning (ML) has been widely used in weather forecasting applications. This popularity stems from its fast inference speed and ability to model complex physical dynamics directly from data. Some high-profile ML weather forecasting models include FourCastNet (Pathak et al., 2022), GraphCast (Lam et al., 2023), and Pangu-Weather (Bi et al., 2023). These ML weather models can generate forecasts thousands of times faster than traditional numerical weather prediction (NWP) models, while at the same time being more accurate and flexible, freed from the NWP model's sometimes restrictive physical constraints (Pathak et al., 2022; Lam et al., 2023; Kochkov et al., 2024).

To date, most ML weather models have been trained with gridded numerical weather reanalysis products like ERA5 (Hersbach et al., 2020). However, reanalysis products have been shown to have a systematic bias relative to the weather station measurements (Ramavajjala & Mitra, 2023). Here, we verify the existence of a substantial bias in ERA5's near-surface wind estimates (Figure 1). ERA5 systematically overestimates the inland near-surface wind speed and is much smoother across space than the actual wind field as measured by weather stations. ML models trained to predict reanalysis products inherit this significant bias and are unable to make accurate localized predictions.

This presents a challenge, as accurate off-grid weather forecasts are critical for applications like wildfire management and sustainable energy generation. To bridge this gap, we train a multi-modal graph neural network (GNN) end-to-end to downscale gridded forecasts to localized off-grid coordinates. First, we curate a dataset that contains both global weather reanalysis (ERA5) and local weather station observations (MADIS), spanning 2019–2023 and covering the Northeastern United States. We then construct a heterogeneous graph containing gridded ERA5 and off-grid weather stations as two different types of nodes. The GNN operates on this graph and makes forecasts at each weather station. It preserves off-grid station nodes' irregular geometry and theoretically infinite spatial resolution. When making predictions at a station location, the GNN aggregates information from neighboring weather stations and ERA5 nodes using message passing (Gilmer et al., 2017). As a result, the prediction is informed by both global atmospheric dynamics and local weather patterns.

We evaluate our model's ability to forecast real data from weather stations, focusing on wind in particular; near-surface wind dynamics are very complex and poorly captured by ERA5. Our GNN method outperforms a variety of other off-grid forecasting methods, including ERA5 interpolation and time series forecasting without spatial context.

Our contributions can be summarized as the following:

- We compile and release a multi-modal weather dataset incorporating both gridded ERA5 and off-grid MADIS weather stations. The dataset covers the Northeastern US from 2019–2023 and includes a comprehensive list of weather variables.

- We verified, using our dataset, the systematic bias between gridded global weather reanalysis product ERA5 and off-grid local weather station measurements.

- We propose a multi-modal GNN to model local weather dynamics at the station level, taking advantage of both ERA5 and weather station observations.

- We evaluate our GNN against a range of data-driven and non-data-driven off-grid weather forecasting methods. Amongst those, our model achieves the best performance. It decreases the average error by 3.57% comparing to the best performing MLP model which reduces the mean wind vector error by 79.34% comparing to interpolated ERA5.

- We conducted an ablation of ERA5 inputs and observed that a GNN with ERA5 nodes achieves 75% of the error of a GNN without ERA5, indicating that—even in the presence of historical station data—global atmospheric dynamics inform local weather patterns.

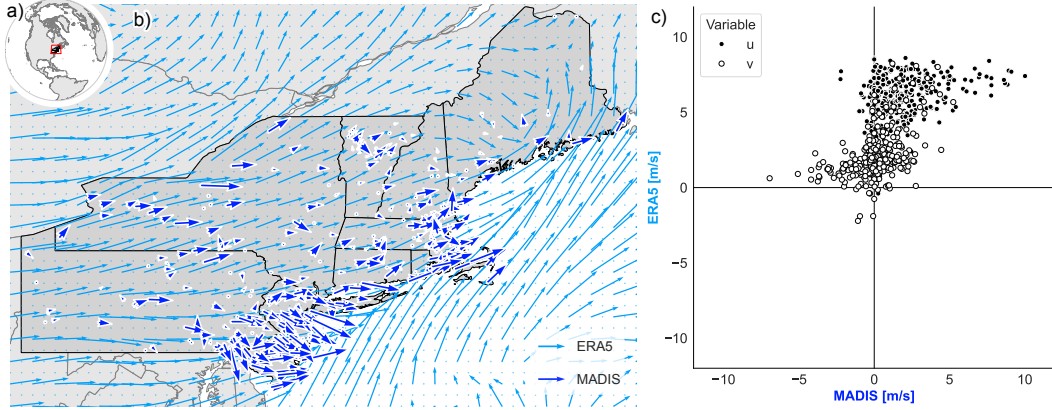

Figure 1: **Gridded reanalysis data like ERA5 do not capture localized, near-surface wind dynamics.** (a) Our study area is the Northeastern United States. (b) Wind field for April 18[th] 2023 18:00–19:00, with ERA5 global reanalysis data in light blue and wind measured by weather stations (MADIS) in dark blue. (c) Scatterplot of MADIS vs. ERA5 for the same time, separated into $u$ and $v$ components of wind. The ERA5 data is linearly interpolated to the locations of the MADIS weather stations.

## 2    RELATED WORK

**Gridded Weather Forecasting**    Weather forecasting has long been a challenging problem in atmospheric sciences, with efforts dating back centuries. Since the advent of numerical weather prediction (NWP) in the mid-20th century, most forecast simulations have been conducted on a regular grid, dividing the atmosphere into evenly spaced discrete points to solve complex partial differential equations. This grid-based approach has remained the foundation of many numerical weather forecasting models such as the Integrated Forecast System (ECMWF, 2022) and High-Resolution Rapid Refresh (Dowell et al., 2022). In recent years, machine learning (ML) has gained traction as a promising tool in weather forecasting (Bauer et al., 2015), offering new techniques to improve accuracy and computational efficiency. These ML weather models can be roughly divided into two

categories: end-to-end models and foundation models. FourCastNet (Pathak et al., 2022), Graph-Cast (Lam et al., 2023), Pangu-Weather (Bi et al., 2023), AIFS (**?**) and NeuralGCM (Kochkov et al., 2024) are end-to-end models trained directly to make weather forecasts. In contrast, At-moRep (Lessig et al., 2023), ClimaX (Nguyen et al., 2023), Aurora (Bodnar et al., 2024) and Prithvi WxC (**?**) are foundation models that are first trained with a self-supervision task and then fine-tuned for weather forecasting. However, the training data for ML models largely stem from traditional gridded numerical simulations such as ERA5 (Hersbach et al., 2020). As a result, the ML models themselves still typically maintain the grid-based paradigm even within their more modern forecasting approach. One major disadvantage of gridded weather forecasting is that it is usually limited by its fixed resolution such that it cannot accurately reflect fine-grained local weather patterns (although efforts towards limited area modeling have recently been made (Oskarsson et al., 2023)). Other works focusing on increasing the forecast resolution (Harder et al., 2023; Yang et al., 2023; Prasad et al., 2024) exist, but their methods are mostly tested on synthetic datasets. Work meant to correct ERA5 forecasts exists (Mouatadid et al., 2023), but focus on sub-seasonal forecast at coarse spatial resolution rather than local weather forecasting. In this work, we propose a multi-modal graph neural network which can effectively downscale gridded weather forecasting to match real-world local weather dynamics.

**Off-Grid Weather Forecasting**  Even though gridded weather forecasting is the main focus of the ML community, there have been several attempts to forecast weather off-grid. Bentsen et al. (2023) applied a GNN to forecast wind speed at 14 irregularly spaced off-shore weather stations, each of which was treated as a node within the graph. The model input is the historical trajectory of weather variables recorded at each station. This work has two limitations: the forecasting region is small, only covering 14 stations, and it only considers a single input modality of station historical measurements. MetNet-3 (**?**) takes another approach to off-grid weather forecasting. It trains a U-Net-like transformer (Ronneberger et al., 2015) model that takes multi-modal inputs including weather station observations, satellite imagery, and assimilation products to predict weather at stations. However, both input and output station data are re-gridded to a high resolution mesh (4 km × 4 km), which distorts the off-grid data's original granularity. To address the aforementioned disadvantages, we construct a multi-modal GNN that makes predictions at raw off-grid locations over 358 stations in the Northeastern US, with both numerical weather simulation and station observation as inputs.

**Graph Neural Network for Physical Simulation**  Graph neural networks (GNNs) are a type of deep learning model designed to operate on data structured as graphs, where entities are represented as nodes and their relationships as edges. GNNs provide flexibility to process data with non-Euclidean structures. A GNN learns to capture relationships between nodes by iteratively passing and aggregating information between neighboring nodes, and updating node representations based on their connections. Recently, GNNs have been widely used in physical system simulation. For example, the 2D Burgers' equation can be effectively solved on both a regular and an irregular mesh with GNNs such as MAgNet (**?**) and MPNN (**?**). Sanchez-Gonzalez et al. (2020) used a GNN to simulate particle dynamics in a wide variety of physical domains, involving fluids, rigid solids, and deformable materials interacting with one another. GraphCast (Lam et al., 2023) even showed that a 3D GNN is capable of simulating a global gridded atmospheric system. These successful use cases of GNNs motivate us to apply a graph network to our task for localized off-grid weather forecasting.

## 3 METHODS

We train a message passing neural network (MPNN, Gilmer et al. (2017); Pfaff et al. (2021)), a type of GNN (Scarselli et al., 2009), to forecast weather at the station level with the aid of global weather predictions. At its core, the method uses past local weather station observations to forecast the weather variables of interest at different lead times into the future. This structure is then augmented by forming a heterogeneous graph with the gridded output of a global weather model (could be NWP or ML) known to provide accurate forecasts globally, but lacking accuracy at fine scales. For instance, global models largely neglect surface friction when modeling wind fields (Figure 1). By integrating global forecasts with localized weather data, we can view the task as a correction of global forecasts rather than forecasting *de novo*; that is, our model aims to correct the global forecast

toward the local reality based on prior local observations. This setup enables our model to achieve accurate off-grid near-surface weather forecasting.

## 3.1 MODEL

The fundamental idea of weather forecasting is to predict the weather at a future time $l\Delta t$ (the lead time), given a set of information:

$$\mathbf{w}(t + l\Delta t) = F(\ldots), \tag{1}$$

where $t$ is the current time, $\mathbf{w}$ a vector of weather observations at $n$ different weather stations ($\mathbf{w} = [w_0, \ldots, w_n]$), and $F$ the function mapping input variables to the forecast. When using local historical data to predict the weather, the function $F$ takes the form:

$$\mathbf{w}(t + l\Delta t) = F(\mathbf{w}(t - b\Delta t : t)), \tag{2}$$

where $b\Delta t$ is the number of past time steps considered, called back hours. This equation thus maps past weather data to future weather data, only considering the local weather stations (Figure 2a).

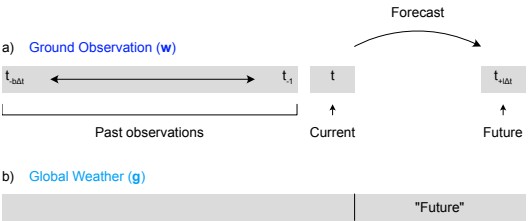

Figure 2: **Schematic of the forecasting task.** (a) The ground stations' past observations $\mathbf{w}(t - b\Delta t : t)$ are used to forecast the weather conditions at a given lead time $\mathbf{w}(t+l\Delta t)$. (b) By introducing a global weather model's past and future data $\mathbf{g}$, the setup is transformed from a pure forecasting problem to a correction problem, where the future global weather data are corrected towards local observations.

We propose to change the nature of the problem, transforming the arguably hard task of forecasting to correcting an existing weather forecast. We thus introduce an external global weather forecast $\mathbf{g}$, and modify the function $F$:

$$\mathbf{w}(t + l\Delta t) = F(\mathbf{w}(t - b\Delta t : t),$$
$$\mathbf{g}(t - b\Delta t : t + l\Delta t)). \tag{3}$$

The global weather forecast covers the period from the back hours all the way to the lead time (Figure 2b). The function $F$ can take the form of any model, for instance a multi-layer perceptron (MLP), a transformer, or, as will be shown here, a GNN, which considers spatial correlation, i.e. the connections between the weather stations, in addition to temporal correlation.

### 3.1.1 MESSAGE PASSING NEURAL NETWORK (MPNN)

We implement the prediction function $F$ as a GNN, which takes advantage of the spatial correlation between off-grid weather stations. Each weather station naturally becomes a node of a graph. The weather station graph is constructed with $k$-nearest neighbor, that is, to connect each node $i$ to a set of nodes $\mathcal{N}(i)$, the $k$ closest neighbors.

MPNNs are a type of GNNs, where messages are passed between connected nodes. The messages consist of information contained in the nodes as well as in the edges connecting the nodes. The nodes are updated with the incoming messages. This architecture can be trained for different tasks, such as predicting at a node level (e.g., simulating particle dynamics) and at a graph level (e.g., classifying chemicals). We follow the implementation of MPNN as described in **?**. It works in three steps: encode, process and decode (Battaglia et al., 2018; Sanchez-Gonzalez et al., 2020).

**Encode** This step encodes the information contained in each node $i$ and transforms it into a latent feature $f$:

$$f_i^0 = \alpha(w_i(t - b\Delta t : t), p_i), \tag{4}$$

where $w$ is a vector containing the observed weather variables at node $i$, $p$ the coordinates, and $\alpha$ an encoding neural network, here a simple two-layer MLP. The superscript of $f$ denotes the number of times the node feature has been processed.

**Process** This step processes each node's feature with incoming messages which aggregate information from its connected neighbors. The node is updated in two phases, iteratively over a total of

$M$ cycles (here $M = 4$):

$$\text{message on edge } j \rightarrow i: \mu_{ij}^m = \beta(f_i^m, f_j^m, w_i(t - b\Delta t : t) - w_j(t - b\Delta t : t), p_i - p_j), \quad (5)$$

$$\text{update node } i: f_i^{m+1} = f_i^m + \gamma\left(f_i^m, \sum_{j \in \mathcal{N}(i)} \mu_{ij}^m/k\right). \quad (6)$$

$\mu_{ij}^m$ is the message, $f_i^{m+1}$ is the updated node feature on the $(m + 1)^{\text{th}}$ processing iteration ($m = [0, \ldots, M - 1]$), and $k$ the number of neighbors in set $\mathcal{N}(i)$. $\beta$ and $\gamma$ are two-layer MLPs.

**Decode**  The decoding step then maps the final node feature to the weather variables at the given lead time:

$$w_i(t + l\Delta t) = \phi(f_i^M), \quad (7)$$

with $\phi$ a two-layer MLP.

### 3.1.2 MULTI-MODAL HETEROGENEOUS GRAPH

To integrate the global weather data (past and future), we propose a multi-modal heterogeneous modeling approach. We first construct a graph that connects the global gridded weather data to the local weather stations (c.f. Figure 3), where each station is paired with its $o$ closest gridded global weather neighbors (4 in the example in Figure 3, but 8 in the experiments later). These edges are uni-directional, meaning the information flows form global to local, but not back. The heterogeneous graph constructed for our study area is given in Figure 5.

To incorporate this new data, we propose to modify the MPNN described above as follows: (1) encode the global data at each node; (2) write a new message passing scheme that propagates the gridded data to the local observations.

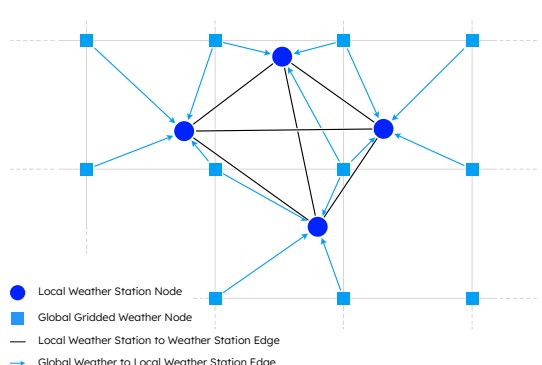

Figure 3: **Simplified diagram of our multi-modal graph.** Local weather stations form the base graph, with each station node connected to its $k$ nearest neighbors. The global reanalysis weather dataset is arranged on a regular grid, with each station node connected to its $o$ closest reanalysis nodes (4 in this example). Station nodes pass messages to each other in bi-directional edges; reanalysis nodes pass messages to station nodes, but not vice-versa.

**Encode Global Node**  The encoding of the global node occurs in a similar way to the one of the local node embedding (eq. 4):

$$h_r = \psi(g_r(t - b\Delta t : t + l\Delta t), p_r), \quad (8)$$

where $h$ is the global node embedding, $\psi$ a two-layer node encoding MLP and $p$ the position.

**Process Global Node**  We then update the embedded local node $f_i^m$ with information aggregated from its $o$ closest global grid nodes $\mathcal{M}(i)$ via message passing:

$$\text{message on edge } r \rightarrow i: \nu_{ir} = \chi(h_r, f_i^m, p_i - p_r), \quad (9)$$

$$\text{update node } i: f_i^{m'} = f_i^m + \omega\left(f_i^m, h_r, \sum_{r \in \mathcal{M}(i)} \nu_{ir}/o\right), \quad (10)$$

where $\chi$ and $\omega$ are two-layer MLPs, and $f_i^{m'}$ the updated local node embedding, which will be substituted in eqs. 5, 6, and 7. We apply this new message passsing scheme to $f^0$ and $f^M$ from the base graph, i.e. (1) after the initial local node encoding (eq. 4) but before the local message passing scheme (eq. 5), and (2) after the last local message update (eq. 6, at iteration $M$), but before the decoding step (eq. 7). One illustration of the model architecture is given in Figure A.2b.

## 3.2 DATA

Our goal in this work is to forecast weather at precise locations that have historical observations. To do this, we use two datasets: (1) point-based weather observations from MADIS stations and (2) gridded reanalysis data from ERA5.

**MADIS** The Meteorological Assimilation Data Ingest System (MADIS[1]) is a database provided by the National Oceanic and Atmospheric Administration (NOAA) that contains meteorological observations from stations covering the entire globe. MADIS ingests data from NOAA and non-NOAA sources, including observation networks from US federal, state, and transportation agencies, universities, volunteer networks, and data from private sectors like airlines as well as public-private partnerships like the Citizen Weather Observer Program. MADIS provides a wide range of weather variables from which we curated 10m wind speed, 10m wind direction, 2m temperature, 2m dewpoint temperature, and surface radiation.

In this work, we focus on stations over the Northeastern US region (Maine, New Hampshire, Vermont, Massachusetts, Rhode Island, Connecticut, New York, New Jersey, and Pennsylvania, see Figure 1a). We only keep averaged hourly observations with the quality flag "Screened" or "Verified". Additionally, only stations with at least 90% of data of sufficient quality are considered. Across the study region, this leaves us with 358 stations (Figure 1a, dark blue arrows). We processed 5 years of data from 2019 to 2023.

**ERA5** The ECMWF Reanalysis v5 (ERA5) climate and weather dataset (Hersbach et al., 2020) is a gridded reanalysis product from the European Center for Medium-Range Weather Forecasts (ECMWF) that combines model data with worldwide observations. The observations are used as boundary conditions for numerical models that then predict various atmospheric variables. ERA5 is available as global hourly data with a $0.25° \times 0.25°$ resolution, which is 31 km/pixel at the equator, spanning 1950–2024. It includes weather both at the surface and at various pressure levels. We curated 5 years (from 2019 to 2023) of surface variables: 10m wind $u$, 10m wind $v$, 2m temperature, 2m dewpoint temperature, and surface radiation.

The details of our curated multi-modal dataset are summarized in Table A.2.

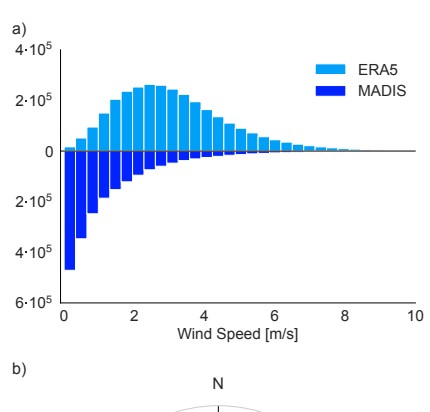

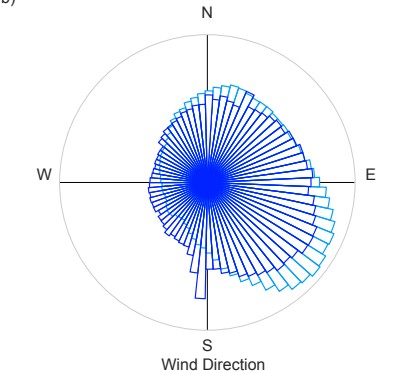

Figure 4: **Comparison of wind data between weather stations and linearly interpolated global reanalysis data.** For both ERA5 and MADIS data, (a) histogram of wind speed and (b) radial histogram of wind direction for the study region from January to December 2023. Large differences, especially in wind speed, are apparent between local wind observations and global wind products. ERA5, which is the target that most ML weather models emulate, does not capture local wind dynamics.

**ERA5 as "Perfect Forecast"** For our task of point-level weather forecasting, we integrate historical station observations with a global forecasting product. At present, most ML weather models are trained to predict ERA5 due to its accuracy, global coverage, relatively high spatial resolution for a gridded global dataset, consistency, comprehensive array of variables, and accessibility (Ibebuchi et al., 2024; Urraca et al., 2018). Rather than choose among the available ML models, we treat ERA5 as the global weather "forecast" input to our model. This way, ERA5 simulates the best-case output of these models and does not introduce an additional forecast error to our method.

**Wind Forecasting** We limit our predictions to wind (speed and direction, expressed as cosine and sine components of the wind vector, i.e., $u$ and $v$) to study the capacity of our modeling approach.

---

[1]https://madis.ncep.noaa.gov/

Due to highly local effects, urban heat islands, and boundary layer complexity from topography, buildings, and trees (Ruel et al., 1998; Auvinen et al., 2020; Liang et al., 2023), near-surface wind is one of the most complicated weather variables to model. Many ML attempts have been made to model wind (Tan et al., 2022; **?**), but the performance is limited. Indeed, the difficulty of modeling near-surface wind can be seen in the discrepancies between ERA5 and ground observations (Figures 1 and 4). ERA5 has a strong tendency to overestimate wind speed over land and fails to capture the effects of local surface elevation on wind direction.

### 3.3 EXPERIMENTS

**Forecast Setup**   To predict the wind vector at each weather station, we first provide the model with 10m wind $u$, 10m wind $v$, and 2m temperature at each MADIS node. Similarly, at each ERA5 grid cell (or node), the inputs are 10m $u$, 10m $v$, and 2m temperature. For all inputs, the temporal resolution is 1 hour. In the MPNN graph, each MADIS node is connected to its 5 nearest MADIS neighbors and its 8 nearest ERA5 neighbors (Figure 5). In early experiments, we tried a fully connected MADIS graph and observed no improvements in performance over 5 nearest neighbors; therefore, to reduce computational cost, we show results for the 5-nearest neighbor graph.

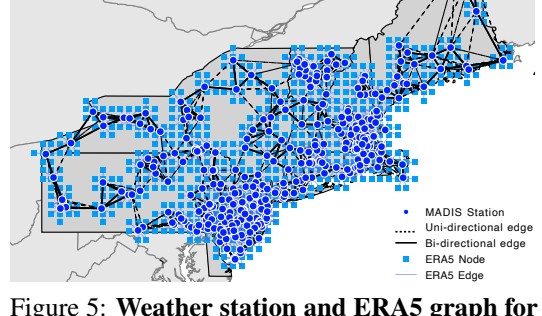

Figure 5: **Weather station and ERA5 graph for the US Northeast study region.** MADIS stations are shown as dark blue circles; the black edges connect each weather station to its 5 nearest neighbors. ERA5 nodes are shown as light blue squares, and each weather station is connected to its 8 nearest ERA5 neighbors.

All baseline and MPNN models are tasked with predicting 10m $u$ and 10m $v$ at each MADIS node for different lead times: 1, 2, 4, 8, 16, 24, 36 and 48 hours, for which we train one model each. The model is trained with data from 2019 to 2021, validated on 2022, and tested on 2023 ($\sim$8,760 time steps per year for each of the 358 stations). The model uses 48 hours of MADIS back hours (Figure 2a), i.e. the weather observations from the previous 48 hours, including the current observation, to predict forward. When including ERA5, the model is given the time steps from the back hours to the lead time (Figure 2b), providing a full temporal view of large scale dynamics.

**Baseline Methods**   We compare the MPNN against a series of baseline forecasting methods (Table A.1): interpolated ERA5, MADIS persistence, and a multi-layer MLP. Interpolated ERA5 refers to a linear interpolation across space of the 8 nearest ERA5 grid cells to a MADIS station location. The MADIS persistence simply shifts the observation by the lead time and will perform well if temporal auto-correlation of wind is high. The MLP provides a baseline model with an architecture mirroring the encoder and decoder of the MPNN, but with no spatial structure (Figure A.2). For the MLP experiments, the ERA5 data is interpolated at the weather stations and used as an additional input. The same MLP is tasked with forecasting at all stations; we also tried training a separate MLP for each station but did not observe better performance.

**Ablation of ERA5**   Both the MLP and the MPNN are run with and without ERA5 data to assess how much performance gain for localized weather forecasting comes from knowing global weather dynamics.

## 4 RESULTS

We report the model performance for the 2023 test set as the mean error (ME) of the predicted wind vector against the actual wind vector ($1/N \sum_i^N \sqrt{\Delta u_i + \Delta v_i}$). Figure 6 and Table A.3 summarize the ME for the different lead times and models. The average ME for each experiment (mean across lead times) is also reported in Table A.3.

**Interpolated ERA5 and persistence both fail to forecast local wind**   Interpolated ERA5 data does not describe the local wind conditions accurately, with an ME at all lead times of 2.71 $m/s$.

(Note that the ME for interpolated ERA5 is constant across lead times, because the "future" values of ERA5 are assumed to be known. This is the best case for a gridded NWP forecast or an ML weather model trained to predict ERA5; if ERA5 were replaced by the output of a global forecasting model, the error would increase with lead time.) Spatially, when considering the ME averaged over lead times for each station (Figure 7), ERA5 appears to consistently misrepresent local weather patterns. The largest magnitude errors are generally concentrated along the coast, due to the higher average wind speeds there, but the highest *relative* errors are inland.

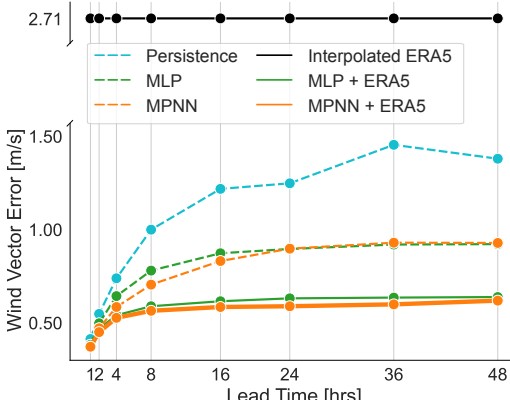

The persistence baseline shows a low ME for short lead times—lead times 1, 2, and 4 hours all have MEs below 0.75 $m/s$. This is expected due to temporal auto-correlation. ME rapidly increases with longer lead times, reaching a maximum ME of 1.46 $m/s$ at 36 hours. Hour 48 shows a slight decrease in ME, hinting at some daily periodicity in wind patterns. Spatially, most of the high ME is also concentrated along the coast, with some exceptions inland. The ME of inland stations is on average substantially lower for the persistence than the interpolated ERA5 baseline.

Figure 6: **Mean error (ME) of the wind vector for different models calculated over all test set time steps and stations, from 1 hour to 48 hours lead times.** The interpolated ERA5 error is calculated by linearly interpolating ERA5 to each weather station; it is the error between the global and local dataset. The persistence is calculated by offsetting the ground truth (MADIS) by the lead time. The results for the multi-layer perceptron (MLP) and message passing neural network (MPNN) are reported with and without the inclusion of ERA5 as input. All methods outperform the ERA5 interpolation, all ML methods outperform the persistence, and the MPNN + ERA5 outperforms all other methods.

**Simple MLP on historical observations outperforms non-ML methods** The MLP + ERA5 method significantly improves on the non-ML methods, showing an ME below 0.65 $m/s$ for all lead times. The MLP error increases with lead hours as expected, reaching the beginning of a plateau at lead hour 16 with an ME of 0.62, hinting towards the model finding a mean value minimizing the forecast error. This can be seen in the MLP prediction flattening out with higher lead times (Figure 8). Overall, the model reduces the ME by 79% compared to the interpolated ERA5, and 44% compared to the MADIS persistence. Spatially, the model improves most along the coast compared to the persistence, better predicting the higher average wind speeds, but also improves on individual stations inland.

**Multi-modal MPNN significantly improves local wind predictions** Using the multi-modal MPNN to correct the global forecast results in an improvement in performance compared to the MLP. The ME is below 0.62 $m/s$ across all lead times and is reduced by 3.6% relative to the MLP. Throughout the different lead times, the MPNN + ERA5 model seems to reach a plateau slightly faster (around lead hour 8), and at a lower ME (0.57 $m/s$), demonstrating a better predictive/corrective power. Along the coast is where the MPNN + ERA5 method shows the strongest improvement in performance: most stations with high errors in the other methods are modeled with a relatively low ME by the MPNN.

**Correcting ERA5 data appears significantly less complex than forecasting with a weather station's historical data.** In our ablation experiments where we remove ERA5 data from the model, both the MLP model and the MPNN perform significantly worse. Both MLP and MPNN show an increase in ME of ∼33% on average, highlighting the significantly harder task of forecasting from the station historical data, compared to correcting the ERA5 data. Figure 8 shows, for the different model, a snippet of time series of wind $u$ and $v$ for the station with the worst ME for the MPNN + ERA5 model. Unlike at many other stations, the interpolated ERA5 data for this location agrees with the local MADIS data well, allowing us to evaluate its integration. The MPNN and MLP model integrate the ERA5 data well, consistently correcting it towards the local prediction. The further out the lead time, the more it relies on the ERA5 data, but still performs better than the interpolated

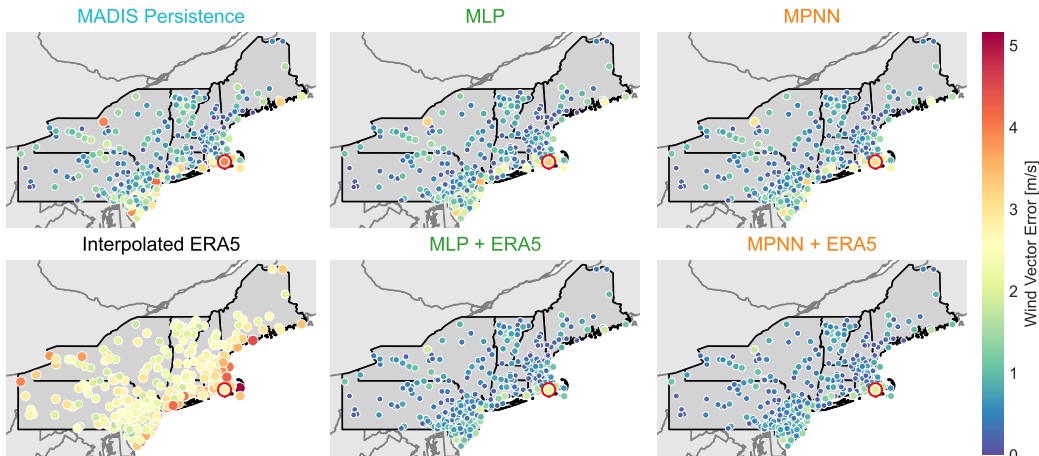

Figure 7: **Average ME of the different methods shown for each MADIS station, averaged over the lead times.** The weather stations along the coast show the highest error and are also where the MPNN + ERA5 model results in the largest improvements over other methods. The station circled in red is the station with the worst ME for the MPNN + ERA5 model, for which a snippet of time series is shown in Figure 8.

ERA5. Meanwhile, without ERA5, the models fails for longer lead times, trending towards a flat wind vector, preferring to minimize error by predicting the average.

**Message-passing improves forecasting ability.** The graph structure improves the ME both with and without ERA5. The introduction of the spatial structure improves the ME by 2.7 and 3.6% (without and with ERA5 respectively). The network density allows neighbors to forward information deeper into the graph.

## 5 DISCUSSION

Discrepancies arise between weather station observations and reanalysis data, such as ERA5, due to the latter's limitations in modeling terrain and land cover. These local surface characteristics significantly influence local weather patterns, particularly wind, but are not adequately represented in gridded global datasets. The high ME associated with MADIS persistence underscores the inherent chaotic nature of wind dynamics; current wind conditions do not reliably indicate future conditions. However, this unpredictability has a less pronounced effect on ME at low wind speeds, typically observed inland and in sheltered coastal regions. Given these complexities, accurate wind forecasting requires a model that accounts for both the discrepancies between global and local conditions and the inherent variability of wind across space and time. ERA5 and historical station data provide complementary information on wind dynamics. We thus in this work seek a method that can integrate these two datasets successfully for forecasting. ERA5 offers large-scale atmospheric patterns, but it fails to capture low wind speeds in areas that are sheltered from wind due to topography or land surface characteristics.

Machine learning methods trained on station data can adapt to local conditions like low wind speeds, but struggle to account for changes between low and high wind speeds (i.e., ramp-ups and ramp-downs) and do not effectively capture the dynamics associated with longer lead times. When predicting local weather using only local data (i.e., in forecasting mode), the MLP and MPNN achieve respectable performances. However, with the introduction of ERA5, the MLP and MPNN, both successfully incorporates large-scale atmospheric dynamics from ERA5, thereby improving predictions at longer lead times. The slightly superior performance of the MPNN suggests that the spatial structure of the GNN, along with the proposed heterogeneous message passing structure, can be suitable for correcting a globally gridded weather dataset to reflect local conditions. Even without ERA5, the inductive bias of the station's spatial correlation within the MPNN does, on its own, enhance forecasting performance.

This study demonstrates the effectiveness of ML methods for integrating diverse data formats in local weather prediction. The proposed GNN model successfully combines the evenly spaced mesh grid of ERA5 data with the irregularly scattered off-grid points of MADIS data.

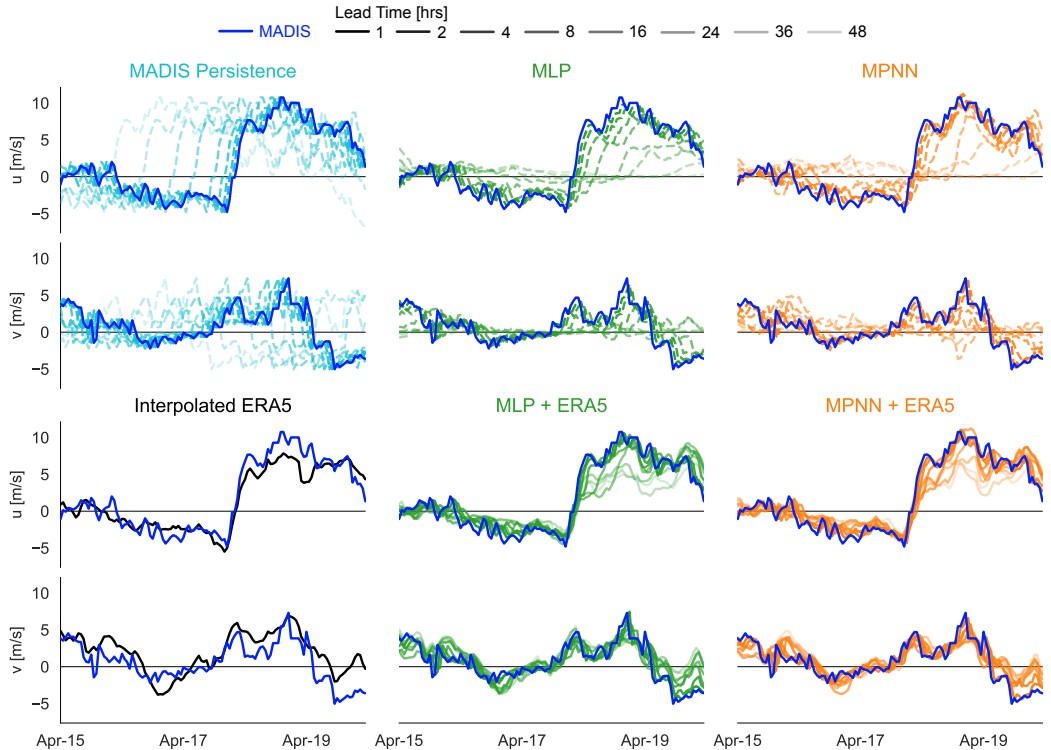

Figure 8: **Example time series from the weather station where MPNN + ERA5 performed worst over all lead times.** The same station is circled in red on the map in Figure 7. For each experiment, the $u$ and $v$ components are shown separately. Each panel has the MADIS ground truth in blue, and the predictions at increasing lead times displayed with decreasing saturation. Interestingly, for this station and time snippet the interpolated ERA5 appears relatively accurate, and the MLP and MPNN are able to take advantage of the ERA5 data. See Figures A.4 and A.5 for further examples of time series, and Figure A.3 for an example of the station's environment.

## 6 CONCLUSION

This work demonstrates the use of a multi-modal GNN for downscaling gridded weather forecasts and improving the accuracy of off-grid predictions. Our model addresses the inherent bias in gridded reanalysis products like ERA5. By incorporating both ERA5 and MADIS weather station data within a heterogeneous graph, our GNN predicts off-grid weather conditions by leveraging both large-scale atmospheric dynamics and local weather patterns. In our evaluation using a surface wind prediction task, the GNN outperformed all baseline models. For instance, it achieved a 80% reduction in ME compared to ERA5 interpolation and a 3.6% improvement over the best-performing multi-Layer perceptron (MLP). An ablation study, where ERA5 input was removed, resulted in an ME increase from 0.54 to 0.72 $m/s$, highlighting the importance of incorporating global atmospheric dynamics for accurate local predictions. This finding motivates the exploration of additional modalities, such as radar measurements and satellite imagery, which could further enhance local forecast accuracy. This research has significant implications for improving weather forecasting, particularly in high-value regions where weather stations can be installed. More accurate off-grid predictions can enhance weather-dependent decision-making in various sectors, including agriculture, wildfire management, transportation, and renewable energy. Future work will focus on expanding the study area and exploring the integration of our GNN model with weather foundation models.

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
