# A APPENDIX

## A.1 EXPERIMENTS

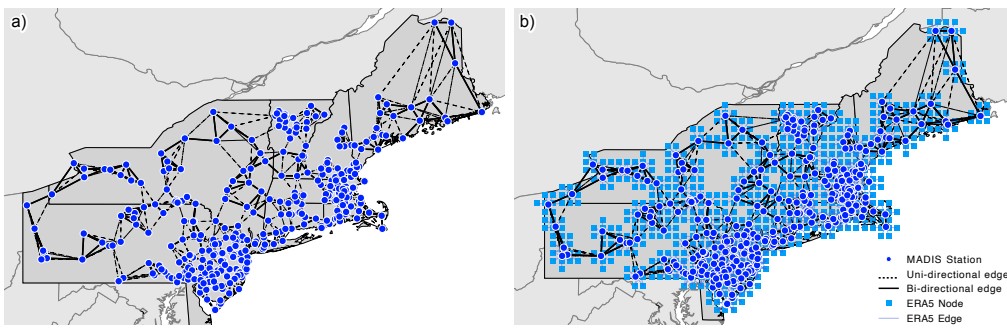

Figure A.1: **Overview of the graph structure for the US Northeast study region.** (a) Graph formed by connecting each of the 358 weather station to its 5 nearest neighbors, and (b) additionally with each of the weather stations connected to the 8 nearest ERA5 nodes, forming the heterogeneous multi-modal graph.

Table A.1: **Summary of experiments.**

| Experiment | ERA5 | ML | Spatial | Multi-modal | Goal |
|---|---|---|---|---|---|
| Persistence | | | | | Similarity to current |
| Interpolated ERA5 | ✓ | | | | Difference to global |
| MLP | | ✓ | | | Forecast |
| MLP + ERA5 | ✓ | ✓ | | ✓ | Correction |
| MPNN | | ✓ | ✓ | | Forecast |
| MPNN + ERA5 | ✓ | ✓ | ✓ | ✓ | Correction |

Our baselines are summarized in Table A.1 and described below.

- The ERA5 data interpolated at the weather stations' location will tell us how accurate global reanalysis data are compared to local observations.
- The MADIS persistence shifts the observation by the lead time and tells us how similar the observation is over time.
- A simple MLP model provides a baseline ML method with no spatial structure.
- The MPNN connects MADIS stations and ERA5 gridded data in a spatial graph.

Both the MLP and the MPNN are run with and without ERA5 future data, and are thus used both to forecast or correct. When ERA5 is not included, both MLP and MPNN only take time series of $u$ and $v$ component of wind and temperature at MADIS stations as input. For MLP, a single model is trained on all MADIS stations simultaneously. For MPNN, it learns on a base graph consisting of only MADIS nodes as in Figure A.1a. The MLP takes interpolated ERA5 at weather stations as input; the MPNN constructs a heterogeneous graph containing both MADIS nodes and ERA5 nodes as shown in Figure A.1b.

## A.2 MODEL ARCHITECTURE

## A.3 DATA

**Processing** We test the methods on the Northeastern United States region (Maine, New Hampshire, Vermont, Massachusetts, Rhode Island, Connecticut, New York, New Jersey, and Pennsylva-

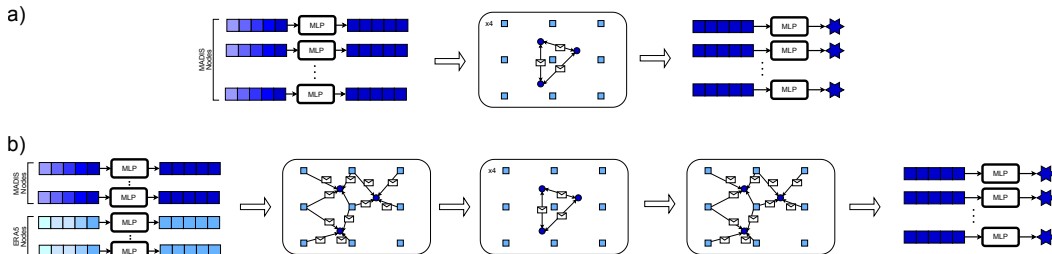

Figure A.2: **MPNN model architecture.** (a) MPNN model architecture without ERA5. Time series of weather variables at MADIS stations are embedded with an MLP. These node embeddings are updated with four iterations of message passing according to a pre-defined graph of MADIS stations. The final node embeddings are decoded with another MLP to generate forecasts at each MADIS station. (b) MPNN model architecture with ERA5. Time series of weather variables at MADIS stations and ERA5 grids are embedded with two separate MLPs. MADIS node embeddings are first updated with messages coming from neighboring ERA5 nodes. They are then iteratively updated four times based on a MADIS station graph (the same graph as (a)). The node embeddings are lastly updated with neighboring ERA5 nodes again; are finally decoded as forecasts by an MLP.

Table A.2: **Summary of our curated dataset.** This consists of three parts: ERA5, MADIS, and ERA5 interpolated to MADIS station locations.

| Name | Type | Temporal Span | Spatial Span | Variables |
|---|---|---|---|---|
| ERA5 | Gridded Mesh | 2019–2023 | Northeast US | 10m $u$, 10m $v$, 2m temperature, 2m dewpoint temperature, surface radiation |
| Interp. ERA5 | Off-Grid Station | 2019–2023 | Northeast US | 10m $u$, 10m $v$, 2m temperature, 2m dewpoint temperature, surface radiation |
| MADIS | Off-Grid Station | 2019–2023 | Northeast US | 10m wind speed, 10m wind direction, 2m temperature, 2m dewpoint temperature, surface radiation |

nia; Figure 1a. The MADIS data is processed for quality, only keeping hourly observations with the quality flag "Screened" or "Verified", and aggregated by hour, taking the mean hourly observation. Additionally, only stations with at least 90% of data of sufficient quality are considered. In the end, over the study regions, it yields 358 stations (c.f. Figure 1 a, dark blue arrows). We select the 10m wind speed $s$ and wind direction $d$ variables, and derive the $u = \cos(270 - d) \cdot s$ and $v = \sin(270 - d) \cdot s$ wind components. We processed 5 years of data (2019 to 2023, included), and split the data in train, validation and test sets, with the train data containing the hourly MADIS data for 2019 to 2021, validation 2022 and test 2023. For ERA5, we use the 10 meters above ground $u$ and $v$ wind components directly provided as is. For certain models (c.f. section 3.3), we linearly interpolate the ERA5 data towards the location of the MADIS weather stations, using the 8 closest ERA5 nodes, inversely weighted by distance.

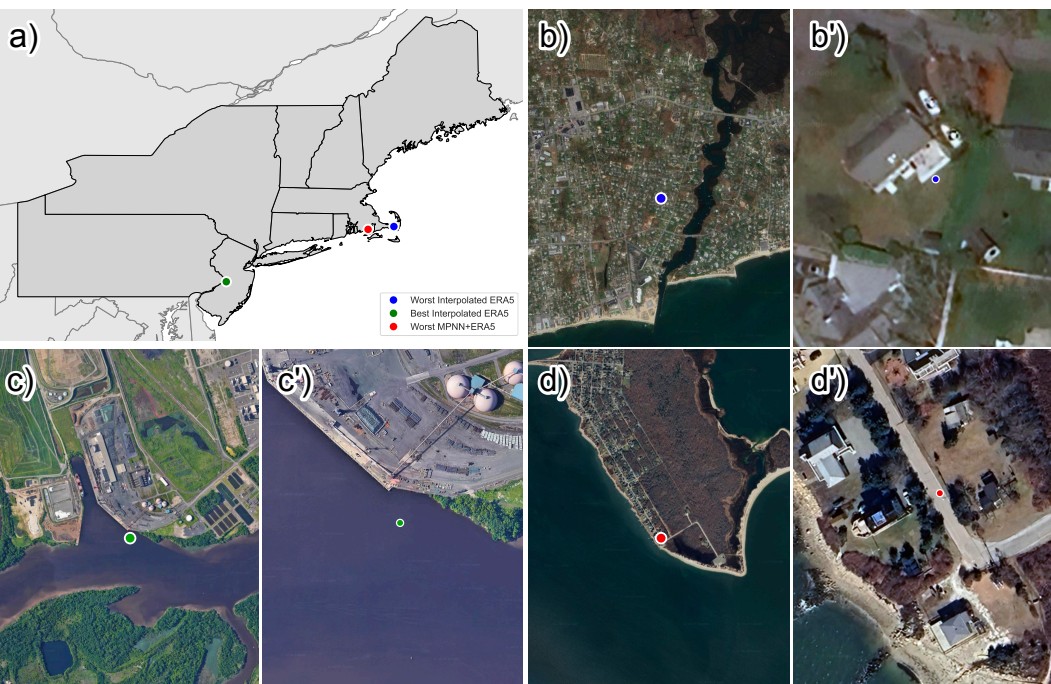

Figure A.3: **Examples of environments for three selected weather stations.** (a) Overview of the three stations. The three selected weather stations are: (b) the station with the worst overall interpolated ERA5 ME (Figure A.4), (c) station with the best overall interpolated ERA5 ME (Figure A.5) and (d) station with the worst overall MPNN + ERA5 ME (Figure 8). (b), (c) and (d) show a zoomed-out view of each area; (b'), (c') and (d') are zoomed in. (b) and (d) show good examples of weather stations being surrounded by trees and buildings, where wind is affected by these local surface characteristics.

## A.4 Stations Environment

## A.5 Additional Results

Table A.3: **Test set Wind Vector Error of experiments for each lead time.** Wind Vector Error is averaged over stations and time steps for the year 2023.

| Model | ERA5? | Lead Time [hrs] | | | | | | | | Average ME |
|---|---|---|---|---|---|---|---|---|---|---|
| | | **1** | **2** | **4** | **8** | **16** | **24** | **36** | **48** | |
| Interpolated ERA5 | ✓ | 2.71 | 2.71 | 2.71 | 2.71 | 2.71 | 2.71 | 2.71 | 2.71 | 2.71 |
| Persistence | ✗ | 0.41 | 0.55 | 0.74 | 1.00 | 1.22 | 1.25 | 1.46 | 1.38 | 1.00 |
| MLP | ✗ | 0.38 | 0.5 | 0.64 | 0.78 | 0.87 | 0.9 | 0.92 | 0.92 | 0.74 |
| MLP + ERA5 | ✓ | 0.38 | 0.47 | 0.54 | 0.59 | 0.62 | 0.63 | 0.64 | 0.64 | 0.56 |
| MPNN | ✗ | 0.37 | 0.47 | 0.59 | 0.71 | 0.83 | 0.9 | 0.93 | 0.93 | 0.72 |
| MPNN + ERA5 | ✓ | **0.37** | **0.45** | **0.53** | **0.57** | **0.59** | **0.59** | **0.6** | **0.62** | **0.54** |

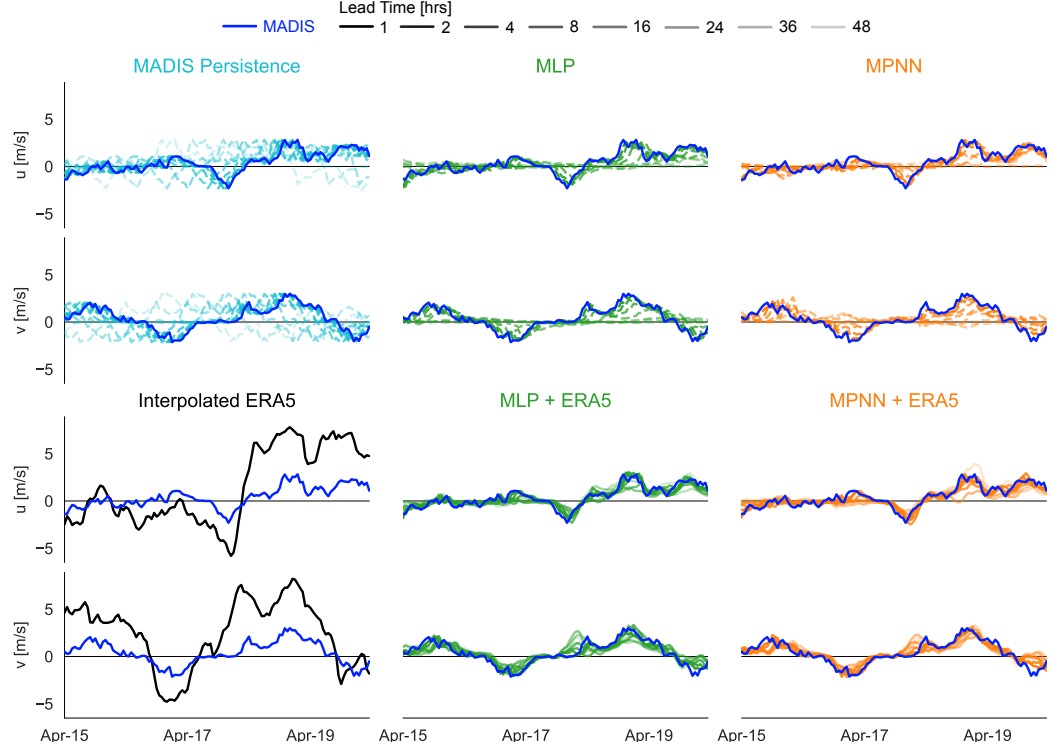

Figure A.4: **Example time series from the weather station where interpolated ERA5 performed worst over all lead times.** This weather station highlights how ERA5 wind magnitude can be quite different from the reality on the ground (c.f. Figure A.3b for an example of station environment). The figure also shows how the MPNN + ERA5 successfully corrects the ERA5 magnitude, but still incorporates it.

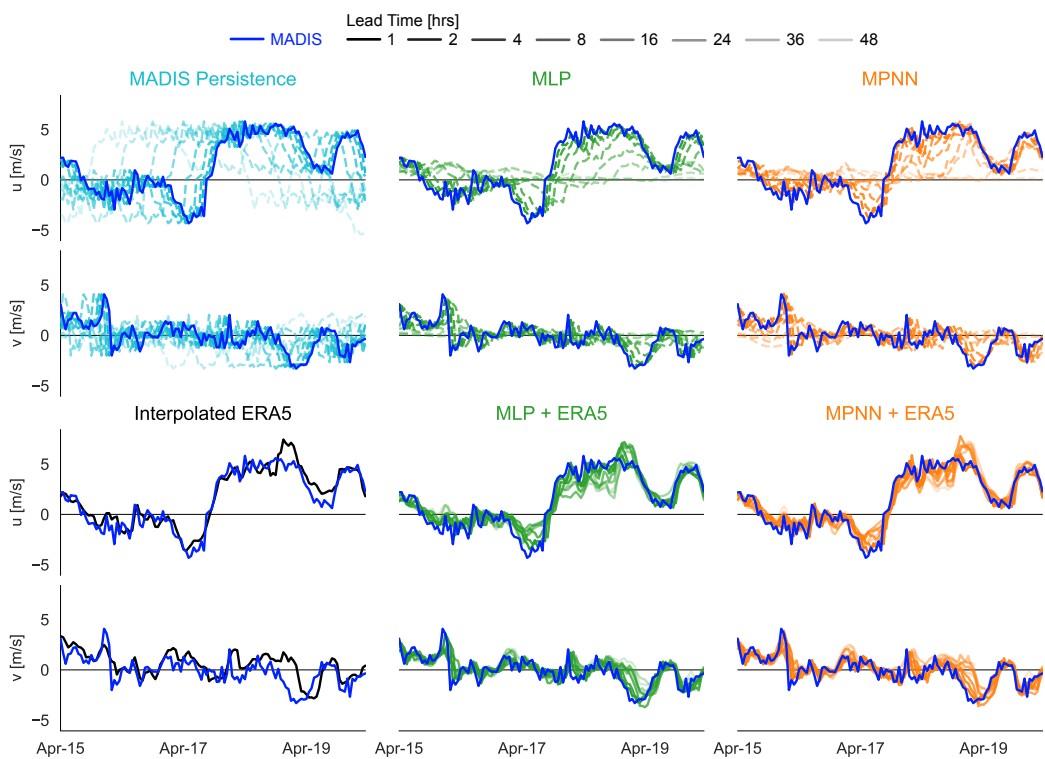

Figure A.5: **Example time series from the weather station where interpolated ERA5 performed best over all lead times.** This weather station highlights how ERA5 can be quite closely matched with a weather station's observations, especially when the weather station is in an open environment (c.f. Figure A.3c for an example of a station in an open environment).