# OpenReview forum: "Multi-modal graph neural networks for localized off-grid weather forecasting"
_ICLR.cc/2025/Conference — Submitted to ICLR 2025_

### Official Review · Reviewer_dgLi · 2024-10-26

**Soundness:** 3
**Presentation:** 2
**Contribution:** 2
**Rating:** 5
**Confidence:** 4

**Summary:**

This paper develops a multi-modal graph neural network (GNN) model to improve localized, off-grid weather forecasting by integrating global and local weather data. The model uses global ERA5 reanalysis data and local MADIS weather station observations, constructing a heterogeneous graph to leverage spatial and temporal correlations, with message-passing neural networks (MPNNs) to predict future wind conditions. It improved prediction accuracy over baseline methods for off-grid weather forecasting.

**Strengths:**

1. The paper has a clear application scenario. The abstract introduces the use case of “precise, localized weather forecasting” right from the start, mentioning scenarios such as “fire management” and “renewable energy generation.”
2. The figures in the paper have a consistent style, with coordinated color schemes and clear content, making the overall presentation visually appealing.
3. The experiments are well-designed with reasonable spatial and temporal ranges. The ablation study investigating the impact of ERA5 inputs on model performance adds to the comprehensiveness of the experiments.

**Weaknesses:**

Firstly, I believe the paper's quality has not yet reached the standard of a top conference like ICLR. I suggest the author consider submitting to a more suitable venue or work towards a more impactful contribution to the topic, rather than an incremental improvement.  I have shown some weakness that I can find in the paper writting aspect but the essential problem of this paper is that lacks breakthrough novelty.

1. The Introduction section lacks a discussion of the motivation. It does not systematically explain the motivation behind the work, nor does it thoroughly compare the proposed model with previous models.
2. The summary of contributions at the end of the Introduction has a high degree of repetition with earlier sections and is not concise or well-summarized.
3. The innovative aspect of the model is restrictive. The paper only uses local observation data to adjust global forecasts, which is restrictive.
4. The presentation of Figure 1 is not closely connected to the context and does not provoke further thought. It is merely shown without being integrated into the discussion. The symbol representing time in Figure 2 is also a bit small.
5. The related work section only briefly lists some relevant works without providing a systematic or organized analysis. Specifically, in the Gridded Weather Forecasting section, only two types of machine learning methods are listed without explaining their advantages or improvements. It also fails to highlight how the proposed model builds on or innovates beyond previous models.
6. The Discussion and Conclusion sections are not concise enough. Some paragraphs are overly lengthy, especially in the experimental results section where certain analyses are repeated, making the content appear redundant.
7. The paper uses many technical terms related to weather forecasting, which may be difficult for cross-disciplinary readers to understand due to a lack of background information.
8. The paper does not mention the conditions for actual deployment or the hardware resources required, making it hard to replicate the results.
9. The paper does not discuss limitations, nor does it analyze the computational costs, which affects the credibility of the work.
10. The Method section lacks an analysis of the model architecture, making it difficult for readers to understand the internal structure of the model.
11. The paper places the model architecture diagram in the Appendix, which makes it inconvenient for readers and easy to overlook. The author should reorganize the content of the paper.
12. The paper only describes the meaning of each term in the equations (eqs. 6 and 10) without explaining the overall logic of the equations. There is no explanation of the internal relationships between the equations. The explanations of equations in the Method section are isolated and not interconnected.
13. The paper's baseline is incomplete and does not include state-of-the-art approaches.

**Questions:**

1. The paper selects the 5 nearest MADIS neighbors and the 8 nearest ERA5 neighbors to construct the graph structure. Why were these numbers chosen? Why did a fully connected MADIS graph observe no improvements? What about a fully connected ERA5 graph?
2. What is the reason for the slight reduction in error in the 48-hour prediction?

---

### Official Review · Reviewer_wNPa · 2024-11-02

**Soundness:** 2
**Presentation:** 3
**Contribution:** 1
**Rating:** 3
**Confidence:** 4

**Summary:**

This paper combines global reanalysis data (ERA5) and local weather station data (MADIS) to forecast the future values of weather stations. They construct a heterogeneous graph to connect the nodes represented by weather stations and grids and achieve the forecasts using a GNN. The proposed model outperforms simple interpolation and persistence methods.

**Strengths:**

1. It is interesting to combine local weather station data and global gridded weather data for localized weather forecasting.
2. It is promising to introduce machine learning methods to address the weather forecasting tasks.

**Weaknesses:**

1. As weather station forecasting is essentially a spatio-temporal prediction task, the introduction of GNNs and heterogeneous graph for this type of problem is not novel that has been extensively explored in prior research [1].
2. The validation is very weak that only a few simple interpolation and persistence methods are compared. Incorporating more powerful numerical weather prediction and machine learning methods would strengthen the evaluation.
3. The absence of physical constraints in the proposed model raises significant concerns about the reliability and robustness of the model.
4. No code and datasets are provided, making this paper difficult to replicate.

[1] Spatio-temporal graph neural networks for predictive learning in urban computing: A survey. TKDE, 2023.

**Questions:**

From Figure 6, while the improvement of introducing ERA5 is limited for MLP, why is it so significant for MPNN?

---

### Official Review · Reviewer_y6WF · 2024-11-03

**Soundness:** 3
**Presentation:** 3
**Contribution:** 2
**Rating:** 5
**Confidence:** 4

**Summary:**

This paper first collected a dataset that contains both global weather reanalysis (ERA5) and local weather station observations (MADIS), spanning 2019–2023 and covering the Northeastern United States. Then this paper applied a heterogeneous graph model on this graph dataset and makes forecasts at each weather station. The proposed dataset off-grid station nodes’ irregular geometry and theoretically infinite spatial resolution. In summary, I consider the major contribution is the propsed dataset. However, this paper lacks in-depth analysis towards the proposed dataset, such as experiments on current SOTA baselines et al.

**Strengths:**

1. This paper compile and release a new multi-modal weather dataset incorporating both gridded ERA5 and off-grid MADIS weather stations. The dataset covers the Northeastern US from 2019–2023 and includes a comprehensive list of weather variables.

2. This paper propose a multi-modal GNN to model local weather dynamics at the station level, taking advantage of both ERA5 and weather station observations.

**Weaknesses:**

1. What innovative aspects does this article's model possess? Is it solely the application of heterogeneous graph networks to weather forecasting?

2. What is the distinction between the proposed dataset and existing datasets is a key contribution of this work？Given that the dataset is the primary innovation, the main text should include a more comprehensive introduction and analysis of its unique characteristics.

3. The paper lacks experiments ( such as experiments on current GNN baselines ), and the volume of experimental work falls short of the acceptance standards required by ICLR.

**Questions:**

Please refer to the weaknesses

---

### Official Review · Reviewer_8RFz · 2024-11-04

**Soundness:** 2
**Presentation:** 3
**Contribution:** 2
**Rating:** 3
**Confidence:** 4

**Summary:**

The authors propose a forecast correction model which using MPNN as a basemodel. They use ERA5 forecasts and weather station data as input to predict wind speed at localised to Northeastern US region.

**Strengths:**

The authors explore the localised forecasting problem here which is indeed a crucial problem, further they treat the problem as multi modality correction problem as compared to only using ERA5 for training a forecast model.

**Weaknesses:**

1. The authors have only used their methodology for the correction of wind forecasts, though they have given the reasons of doing this weather forecasts collectively depends on a range of variables and it would be good to test their methodology on other weather variables as well.
2. In experiments MPNN is mostly compared with MLP only (which is bound to improve results), some of the SOTA forecasting models are based on Vision Transformers and Diffusion models which should be explored and added in the comparison here.
3. Reproducibility parameters are missing

**Questions:**

1. What is the computational complexity of training MPNN on multi-modal data?
2. In the paper authors have mentioned "our approach improves predictions at longer lead times". what does longer lead time mean here as the results are shown for upto 48 hours only. How does the model performs on longer lead times say 7 to 14 days?
3. Lacks training details for the proposed approach as well as baselines used in comparison. Also the training runtimes and number of GPUs and devices are missing.

---

### Official Review · Reviewer_Ugv9 · 2024-11-04

**Soundness:** 2
**Presentation:** 3
**Contribution:** 2
**Rating:** 5
**Confidence:** 5

**Summary:**

The paper proposed a multi-model GNN approach to fuse the global-level ERA5 weather data and station-level observation data for accurate off-grid predictions, the idea is interesting and the paper is written well to follow.

**Strengths:**

1. the problem is critical and the proposal idea is interesting, namely correction instead of pure predictions may help for the off-grid forecasting
2. The paper is easy to follow and the results are reasonable

**Weaknesses:**

1. While the intuition is reasonable, the proposed solution is confused. It seems like a post-calibration method that widely used in the community. For example, instead of doing the ERA5 prediction as proposed in the paper, the prediction results can be further calibrated with the off-grid observations, in this way, the accuracy will be greatly improved. In this case, what is the difference between the proposed approach and the above method. I also would like to see the results.
2. Missing baselines. There are many SOTA baselines for either grid-based predictions or off-grid predictions, but the authors only compare with some basic methods, more baselines are needed to validate the effectiveness of the proposed approach.
3. Since the method rely on the forecasting results of ERA5, the accuracy will be affected heavily by its values. More experiments and discussion should be added.

**Questions:**

See the weakness

---

> ### Author Response · Authors · 2024-11-15
> **References regarding your review**
>
> Thank you for your detailed feedback. We would appreciate some additional clarification to help us better address your concerns and improve our manuscript.
>
> Regarding your first point about post-calibration methods:
> - Could you please provide specific references to the post-calibration methods you mentioned that are widely used in the community? This would help us better understand the similarities and differences with our approach, and enable us to provide a more thorough comparison.
> - We are particularly interested in papers that demonstrate the ERA5 prediction calibration method you described using off-grid observations.
>
> Concerning your second point about missing baselines:
> - We would be grateful if you could point us to the specific SOTA baselines you believe should be included in our comparison for off-grid predictions.
> - This would ensure we can conduct a comprehensive evaluation against the most relevant and current methods in the field.

---

### Meta-Review · Area_Chair_8VU5 · 2024-12-20

**Metareview:**

This paper presents a multimodal graph neural network, specifically MPNN-based, pretrained on global ERA5 dataset and off-grid weather stations (MADIS) for downscaling. The target is to predict wind conditions at several local stations.
The paper is well written.
However, the paper is missing many comparisons with SOTA baselines especially GNN based methods for climate and weather forecasts, and for downscaling.  The comparison is only done with MLP.
Considering the many gaps remaining, this paper is not yet ready for publication.

**Additional Comments On Reviewer Discussion:**

There was no further discussion as the authors only replied to one reviewer.

---

### Decision · Program_Chairs · 2025-01-22

Reject